# A Novel Multi-Gene Combined RT-PCR Assay for Rapid and Sensitive Detection of Maize Dwarf Mosaic Virus

**DOI:** 10.3390/v17030370

**Published:** 2025-03-05

**Authors:** Yujie Jin, Xihong Chen, Min Li, Xiaoqi Zhang, Wei Cai, Jianguo Shen, Yongjiang Zhang, Fangluan Gao

**Affiliations:** 1Institute of Plant Virology, Fujian Agriculture and Forestry University, Fuzhou 350002, China; jinyujie2333@163.com (Y.J.); zhangxiaoqi202202@163.com (X.Z.); 2Fujian Key Laboratory for Technology Research of Inspection and Quarantine, Technology Center of Fuzhou Customs District, Fuzhou 350001, China; kaishilvxing00@163.com (X.C.); limin_2926@163.com (M.L.); greatcai@126.com (W.C.); 3Chinese Academy of Inspection and Quarantine, Beijing 100121, China

**Keywords:** maize dwarf mosaic virus, multi-gene combined RT-PCR assay, real-time quantitative PCR detection (RT-qPCR), molecular detection

## Abstract

Maize, a staple food and cash crop worldwide, also serves as a critical industrial raw material. However, it is significantly threatened by viral pathogens, particularly maize dwarf mosaic virus (MDMV), the primary cause of maize dwarf mosaic disease, a debilitating condition affecting maize cultivation. This study aims to establish a multi-gene combined RT-PCR assay for the rapid specific, sensitive, and reliable detection of MDMV without the need for special expensive equipment. Samples of imported maize, sorghum, and barley were collected from ports in Fujian and Shanghai. Primers targeting the coat protein (CP) and cytoplasmic inclusion protein (CI) genes of MDMV were designed and optimized. Through the design and screening of primers, as well as the optimization of reaction conditions and primer concentrations, a multi-gene combined RT-PCR assay was established to simultaneously detect both genes. Additionally, a real-time fluorescent-based RT-PCR (RT-qPCR) assay was developed using the CP gene to confirm the accuracy of multi-gene combined RT-PCR assay. The sensitivity of the optimized multi-gene combined RT-PCR assay enables the detection of MDMV in infected maize leaf crude extracts at dilutions of 5.37 pg/μL. This assay exhibited excellent specificity, high sensitivity, and robust repeatability, providing swift and accurate detection of MDMV. The multi-gene combined RT-PCR assay offers precise and efficient technical support for MDMV detection and contributes to improved maize production practices.

## 1. Introduction

Maize is one of the world’s most important food and cash crops, as well as a key industrial raw material, cultivated extensively due to its considerable economic value. However, maize viral diseases pose a serious threat to global production [1]. To date, over 40 distinct types of maize viruses have been reported worldwide [2], significantly restricting the development of the maize industry [3]. Of those, maize dwarf mosaic is particularly noteworthy, manifesting through symptoms such as dwarfing, chlorosis, and mosaic patterns on leaves. The primary causative agent of this disease is maize dwarf mosaic virus (MDMV), a single-stranded, positive-sense RNA virus belonging to the *Potyvirus* genus. It is transmitted by aphids, rust spores, and seeds [4]. Notably, the virus exhibits a relatively narrow host range, infecting only a small number of gramineous crops, such as maize and sorghum. MDMV is prevalent in maize-growing regions across Europe, America, Africa, and Australia. Recent studies suggested that its infection range is progressively expanding on a global scale [1,5,6]. However, recent studies conducted by Chinese researchers indicated that the causative agent of maize dwarf mosaic disease in China is sugarcane mosaic virus (SCMV) rather than MDMV [7]. China has classified MDMV as a quarantine pest under the List of Quarantine Pests of Imported Plants. During early infection, MDMV mainly impacts the leaves, leading to yellow streaks, chlorosis, and varying degrees of mosaic patterns. This is very similar to the symptoms of other major viruses that infect maize. As the disease progresses, affected plants exhibit stunted growth, reduced heading rates, lower pollen viability, and poor ear development [2]. The virus not only infects maize independently but can also co-infect with maize chlorotic mottle virus (MCMV), wheat streak mosaic virus (WSMV), and SCMV, resulting in maize lethal necrosis (MLN). This condition poses a significant threat to production stability and the economic viability of maize-growing regions worldwide [8].

Currently, detection of plant viruses is typically conducted using methods such as indicator plant identification, electron microscopy, serological assays, molecular biology techniques, and high-throughput sequencing [9,10,11,12,13]. However, during the quarantine process, traditional methods like indicator plant identification and electron microscopy for detecting MDMV are both time-intensive and labor-intensive. While serological methods, such as ELISA, enable the simultaneous testing of a large number of samples, they are susceptible to cross-reactivity [14]. Molecular techniques for MDMV detection, including RT-PCR, digital RT-PCR [15], RT-LAMP [16], and next-generation sequencing (NGS) technologies [17], have been developed to address these challenges. Notably, NGS offers exceptional accuracy but remains economically unfeasible for routine diagnostic use due to its high cost [18].

Mutations, genetic drift, and selective pressures can cause viruses, especially RNA viruses, to undergo a variety of sequence variations. As a member of the *Potyvirus* genus, MDMV shares significant genetic similarity with other potyviruses in the same group [19,20], and as an RNA virus, it has high mutation rate [2], which complicates accurate identification through single serological assays or conventional PCR methods. To address this challenge, a multi-gene combined detection technology has been developed, targeting multiple viral gene fragments [21].

The multi-gene combined RT-PCR assay enables the simultaneous amplification of the conserved CP and CI genes of MDMV in a single reaction system. By targeting both genes concurrently, it overcomes the issue of false-negative results that are often caused by variations in individual gene fragments. This dual amplification strategy improved the accuracy and reliability of detection, facilitating more precise and rapid identification of MDMV. In this study, an RT-qPCR assay based on the CP gene was also developed to validate the practical application of the MDMV multi-gene combined RT-PCR assay. Collectively, these efforts provide essential technical support for maize quarantine measures and practical applications in agricultural production.

## 2. Materials and Methods

### 2.1. Experimental Material

The analyzed samples included maize, sorghum, and barley seeds, all of which were collected from the ports of Fuzhou, Xiamen, and Shanghai in China (Appendix A). Positive control samples for MDMV, SCMV, Johnson grass mosaic virus (JGMV), WSMV, MCMV, barley stripe mosaic virus (BSMV), tobacco mosaic virus (TMV), southern rice black stripe dwarf virus (SRBSDV), and cucumber mosaic virus (CMV) were maintained by the Technology Center of Fuzhou Customs District.

The complete genome sequences of the selected representative MDMV isolates (detailed accession numbers in Appendix A) were retrieved from the NCBI GenBank database and used for primer design. Primers for multi-gene combined RT-PCR assay were derived from the conserved CP and CI genes of MDMV, whereas primers and probes for RT-qPCR were specifically designed using sequences only from the CP gene (Table 1). The design parameters were rigorously assessed using Oligo Calc [22], with exclusion criteria based on Gibbs free energy changes (∆G) of <−8 kcal mol^−1^ for hairpin structures and <−9 kcal mol^−1^ for self-dimers and cross-dimers. Conventional RT-PCR was conducted using primers as described by Chen [23]. All the primers were synthesized by Shanghai Sangon Biological Engineering Technology and Service Co., Ltd. (Shanghai, China).

### 2.2. Conventional RT-PCR

RNA extraction was conducted using the E.Z.N.A.^®^ Plant RNA Kit (Omega Bio-tek, Norcross, GA, USA), according to the manufacturer’s protocol, with 0.1 g plant seed powder as the starting material. The RNA concentration was measured by an ultraviolet spectrophotometer (Implen N60, Munich, Germany). cDNA synthesis was carried out through reverse transcription using the StarScript III All-in-one RT Mix with a gDNA Remover kit (GenStar, Beijing, China). For cDNA synthesis, ≤1 μg of total plant RNA was combined with a reaction mixture containing 1 μL of No RT Control Mix, 4 μL of 5× StarScript III All-in-one RT Buffer, and nuclease-free water (DEPC-treated) to a final volume of 20 μL, as per the manufacturer’s guidelines. All experimental steps were performed on ice. The reverse transcription reaction was performed using a PCR apparatus (Biometra T advanced 96SG, Analytik Jena AG, Munich, Germany, the same instrument used in subsequent experiments) under the following conditions: 37 °C for 2 min, 50 °C for 15 min, and 85 °C for 2 min. PCR amplification was then conducted in a 25 μL reaction system consisting of 2 μL of cDNA, 12.5 μL of Go Taq^®^ Green Master Mix (Promega, Beijing, China), 0.4 μmol/L each of the MDMV-482-F and MDMV-482-R primers, and RNase-free ddH_2_O to a final volume of 25 μL. The thermal cycling conditions were as follows: an initial denaturation at 94 °C for 3 min, followed by 35 cycles of 94 °C for 30 s, 49 °C for 45 s, and 72 °C for 1 min, with a final extension at 72 °C for 10 min.

### 2.3. Multi-Gene Combined RT-PCR

By screening four pairs of primers, a multi-gene combined RT-PCR assay for MDMV was established using cDNA from MDMV-positive samples as the template. PCR amplification of each of the four pairs of designed primer pairs was performed under the same experimental conditions. The PCR amplification was carried out in a 25 μL reaction system consisting of 2 μL of cDNA, 12.5 μL of Go Taq^®^ Green Master Mix, 0.4 μmol/L each of the primers, and RNase-free ddH_2_O to a final volume of 25 μL. The thermal cycling conditions were as follows: an initial denaturation at 94 °C for 3 min, followed by 35 cycles of 94 °C for 30 s, 51 °C for 45 s, and elongation at 72 °C for 1 min, with a final extension at 72 °C for 10 min. Meanwhile, pairwise grouping of primers was performed according to the targeted segments (CP-343 and CI-640; CP-343 and CI-490). For these pairwise reactions, the 25 μL reaction system included 2 μL cDNA, 12.5 μL 2× Multiplex buffer (high specificity), 1 μL Multiplex DNA polymerase (Vazyme, Jiangsu, China, high specificity, 10 U/μL), 0.4 μmol/L each of the two primers, and RNase-free ddH_2_O to a final volume of 25 μL. The thermal cycling conditions were as follows: initial denaturation at 95 °C for 5 min, followed by 35 cycles of 95 °C for 30 s, 49 °C for 90 s, and elongation at 72 °C for 1 min, with a final extension at 72 °C for 10 min. The reaction procedure and system were optimized based on electrophoretic analysis. First, the primer combination that produced amplicons of similar intensity was selected for further analysis. Next, the annealing temperature was fine-tuned by calculating the Tm values of the CP and CI gene primers. A temperature gradient, increasing by 2 °C increments from 45 °C to 61 °C, was tested to identify the optimal annealing temperature that maximized the intensity of the two target bands. Finally, following the determination of the optimal annealing temperature, the concentrations of the CP and CI primers were systematically adjusted. Eight different primer concentrations were tested (Appendix A), ensuring an optimal balance between band intensity and size, ultimately leading to the selection of the most effective primer combination.

To rigorously assess the sensitivity of the newly optimized multi-gene combined RT-PCR assay, cDNA from the MDMV-positive sample underwent a series of ten-fold serial dilution. This was achieved by adding 45 μL of RNase-free ddH_2_O to 5 μL of the stock solution at each step, starting from the original concentration down to a 10^−7^ dilution. This approach allowed for a direct comparison of the sensitivity of the multi-gene combined RT-PCR assay with that of conventional RT-PCR. For a thorough evaluation of specificity, a range of cDNA samples negative for MDMV but positive for other viruses, namely SCMV, JGMV, WSMV, MCMV, BSMV, TMV, SRBSDV, and CMV, were used as templates, and the performance of the assay was compared to that of conventional RT-PCR. Lastly, the assay’s repeatability was validated by subjecting the same MDMV-positive sample to five independent replicates under identical experimental conditions.

### 2.4. RT-qPCR Detection

RNA extracted from MDMV-positive samples was used as a template, along with the primers MDMV-CP-343-F/R, in combination with the HiScript^®^ II One Step RT-PCR Kit (Dye Plus, Vazyme, Jiangsu, China). RT-PCR was conducted in a 25 μL reaction mixture consisting of 12.5 μL of 2× One Step Mix (Dye Plus), 1.25 μL of One Step Enzyme Mix, 0.4 μmol/L each of the forward and reverse primers, 1 pg~1 μg of total RNA, and RNase-free ddH_2_O to a final volume of 25 μL. The thermal profile included a reverse transcription at 50 °C for 30 min and an initial denaturation at 94 °C for 3 min, followed by 35 cycles of 94 °C for 30 s, 49 °C for 30 s, and 72 °C for 30 s, with a 7 min extension at 72 °C. Following the completion of PCR, the amplified product was ligated into the cloning vector using the Ultra-Universal TOPO cloning kit (Vazyme, Jiangsu, China) and subsequently transformed into *Mach1 T1R* strain *E. coli* competent cells (AngYu Bio, Shanghai, China). Blue–white screening was carried out to identify successful clones, and the recombinant plasmid was submitted to Beijing Tsingke Biotech Co., Ltd. (Beijing, China) for sequencing analysis.

The plasmid concentration was determined and used as a standard for subsequent experiments. The number of plasmid copies was calculated using the formula: (copies/μL) = [6.02 × 10^23^ × (plasmid concentration (ng/μL) × 10−^9^)]/[plasmid length (bp) × 660].

For RT-qPCR, the Go Taq^®^ Probe qPCR Master Mix Kit (Promega, Beijing, China) was used under light-avoiding conditions in a 25 μL reaction system consisting of 12.5 μL of Go Taq^®^ Probe qPCR Master Mix (2×), 0.4 μmol/L each of MDMV-CP-F/R/P, 2 μL of plasmid, and RNase-free ddH_2_O to a final volume of 25 μL. The experiments were performed using a fluorescence quantitative PCR apparatus (ABI StepOnePlus, Applied Biosystems, Foster City, CA, USA). The thermal cycling conditions were as follows: 95 °C for 2 min, followed by 40 cycles of 95 °C for 15 s and 60 °C for 1 min. The plasmid standard stock solution was serially diluted ten-fold by adding 45 μL RNase-free ddH_2_O to 5 μL stock solution at each step, from its original concentration down to a 10^−9^ dilution. RT-qPCR reactions were performed in triplicate for each concentration. Following the reaction, five consecutive gradient concentrations were selected to establish the standard curve, where the Cycle threshold (Ct) values were plotted on the *y*-axis, and the plasmid standard concentrations were plotted on the *x*-axis. Subsequently, cDNA from the MDMV-positive sample was serially diluted ten-fold by adding 45 μL RNase-free ddH_2_O to 5 μL stock solution at each step, from the original concentration to a 10^−8^ dilution, to assess the sensitivity of the RT-qPCR detection assay. To evaluate the assay’s specificity, a range of cDNA samples negative for MDMV but positive for SCMV, JGMV, WSMV, MCMV, BSMV, TMV, SRBSDV, and CMV were used as templates. For repeatability validation, five different concentration gradients of cDNA from the MDMV-positive sample were tested, with three replicates at each concentration. The mean Ct value, along with the standard deviation and coefficient of variation, was calculated to assess the consistency of the assay results.

### 2.5. Practical Application for Different Methods to Detect MDMV

To evaluate the practical utility of the multi-gene combined RT-PCR assay, it was applied alongside the RT-qPCR detection assay established in this study, as well as conventional RT-PCR, to detect MDMV in maize, barley, and sorghum samples entering the port. The results obtained from the three assays were compared to assess the practical effectiveness of the multi-gene combined RT-PCR assay for applications such as quarantine and production monitoring of MDMV.

## 3. Results

### 3.1. Conventional RT-PCR Detection

In the conventional RT-PCR detection of MDMV, both the sample and the positive control successfully amplified the target fragment of approximately 482 bp, while neither the negative control nor the blank control exhibited any amplification (Appendix A). These results were consistent with expectations.

### 3.2. Multi-Gene Combined RT-PCR

The results of MDMV multi-gene combined RT-PCR assay showed that primers designed based on the CI and CP genes of MDMV could successfully amplify the corresponding fragments separately. Gel electrophoresis results showed that the two primers (MDMV-CP-343-F/MDMV-CP-343-R and MDMV-CI-490-F/MDMV-CI-490-R), when combined to detect simultaneously MDMV, produced the best amplification results. These amplification results were consistent with those obtained from the conventional RT-PCR, with the expected gene fragments being clearly and specifically amplified, and no non-specific amplification observed (Appendix A).

Based on the gel electrophoresis results, the system produced two distinct and specific target bands at annealing temperatures of 45 °C, 47 °C, 49 °C, 51 °C, 53 °C, 55 °C, 57 °C, and 59 °C. However, at 61 °C, only the 343 bp band was amplified, while the 490 bp band was not observed. Notably, at 47 °C, the two gene bands were clear, specific, and exhibited appropriate intensity (Appendix A). Furthermore, at this temperature, the G2 primer dosage combination yielded the best results, with the intensity and size of the two amplified gene bands being nearly identical (Appendix A). Therefore, after optimizing the reaction system, the final optimized conditions were determined as follows: cDNA 2 μL, 12.5 μL 2× Multiplex buffer (High specificity), 1 μL Multiplex DNA Polymerase (High specificity, 10 U/μL), 0.2 μmol/L each of MDMV-CP-343-F/R, 0.8 μmol/L each of MDMV-CI-490-F/R, and RNase-free ddH_2_O to a final volume of 25 μL. The thermal cycling conditions were set as follows: 95 °C for 5 min, followed by 35 cycles of 95 °C for 30 s, 47 °C for 90 s, and 72 °C for 1 min, with a final extension at 72 °C for 7 min.

The results of the sensitivity measurement indicated that the optimized multi-gene combined RT-PCR assay exhibited a high level of sensitivity, successfully detecting MDMV cDNA at dilutions as low as 10^−4^ (Figure 1(a-1)). When MDMV-CI-490 was used for independent detection, the lowest detectable dilution was 10^−5^ (Figure 1(a-2)), while for MDMV-CP-343, the lowest detectable dilution was 10^−3^ (Figure 1(a-3)). In comparison, the sensitivity of conventional RT-PCR was also evaluated, and the results showed that MDMV cDNA could only be successfully detected up to a dilution of 10^−1^ (Figure 1(a-4)). The concentration of the RNA stock solution was measured at 53.7 ng/μL, which indicates that the lowest detection limit of the optimized multi-gene combined RT-PCR assay was 5.37 pg/μL. These findings confirmed that the multi-gene combined RT-PCR assay has high sensitivity.

The specificity results demonstrated that the established multi-gene combined RT-PCR assay could simultaneously amplify both target genes from MDMV-positive maize samples without any non-specific amplification. Additionally, no corresponding bands were amplified in samples containing SCMV, JGMV, WSMV, MCMV, BSMV, TMV, SRBSDV, and CMV (Figure 1(b,b-2)). Sequencing of the PCR products obtained from multi-gene combined RT-PCR assay revealed that the sequences shared more than 93% similarity with the reported MDMV sequences, further confirming the specificity of the multi-gene combined RT-PCR assay.

The repeatability results showed that all five repeated tests produced consistent results, with both target fragments being amplified as expected (Figure 1c). Thus, the multi-gene combined RT-PCR assay exhibited excellent repeatability.

### 3.3. RT-qPCR Detection of MDMV

The standard plasmid concentration ranged from 4.3 × 10^5^ copies/µL to 4.3 × 10^9^ copies/µL. A strong linear relationship was observed between the Ct mean value and the plasmid standard concentration. The slope of the standard curve was −3.34, with a coefficient of determination (R^2^) of 0.997, indicating a PCR amplification efficiency of 99%. The linear equation was determined to be *y* = −3.338*x* + 45.328, where *y* represents the Ct value, and *x* is the logarithmic value of the plasmid concentration (Figure 2a).

The result of the sensitivity test revealed that the Ct values for the MDMV real-time fluorescent PCR detection method at gradient dilutions from 10^−1^ to 10^−5^ were 13.26, 16.71, 19.44, 23.04, and 26.79, respectively. No amplification was observed for dilutions from 10^−6^ to 10^−9^ (Figure 2b). Therefore, the lowest detection limit of the real-time fluorescent RT-PCR method was 0.537 pg/μL.

The result of the specificity test showed that only the MDMV-positive samples produced typical amplification curves, while samples infected with other viruses did not exhibit any amplification. This indicates that the MDMV primers and probes demonstrated strong specificity (Figure 2c).

The result of the repeatability test (Table 2) expressed that the coefficient of variation in Ct values between the calculated five repetitions ranged from 2.01% to 4.18%, both of which were less than 5%, indicating that the established RT-qPCR had good repeatability.

### 3.4. Practical Application for Different Methods to Detect MDMV

Using the optimized multi-gene combined RT-PCR assay, real-time fluorescent PCR assay, and conventional RT-PCR, sixty different samples from the ports of Fuzhou, Xiamen, and Shanghai were tested.

The results of the multi-gene combined RT-PCR assay showed that, among the 60 samples, 8 samples amplified two specific bands of approximately 343 bp and 490 bp, while no target bands were amplified in the remaining 52 samples (Appendix A). The results of RT-qPCR assay were consistent with those of the multi-gene combined RT-PCR assay, with 8 samples showing typical amplification curves, while no amplification was observed in the other 52 samples, thereby confirming the findings of the multi-gene combined RT-PCR assay (Appendix A). However, the results of conventional RT-PCR detection differed slightly, with 7 out of 60 samples amplifying a 482 bp band, while no specific bands of approximately 482 bp were observed in the remaining 53 samples (Appendix A). Eight PCR products derived from MDMV-positive samples were sequenced. The comparison results unequivocally confirmed the presence of MDMV in these samples (Appendix A).

## 4. Discussion

As a quarantine pest, MDMV poses a significant threat to various graminaceous crops, particularly maize. Consequently, limiting the spread of MDMV remains the most effective strategy for controlling this virus.

Currently, PCR technology is widely used for the diagnosis of plant viruses. However, conventional PCR methods primarily target a single viral gene, which is more effective for viruses with well-defined strain classifications. For viruses in the same genus as potato virus Y (PVY), such as CSV, JGMV, SCMV, and MDMV [24], their high genetic similarity makes accurate differentiation using traditional single-gene PCR techniques challenging. This challenge is further compounded by the fact that MDMV is a highly mutable RNA virus [2]. Since a single gene fragment represents only a portion of the viral genome, significant mutations in this segment can hinder accurate detection and lead to false-negative results. This limitation presents significant obstacles in the quarantine of agricultural products in global trade. Multi-gene combined RT-PCR is a specialized form of multiplex RT-PCR, in which multiple pairs of primers are used simultaneously in one reaction system to amplify different gene fragments of the same virus. Previous studies demonstrate that the CP gene in potyvirus genomes is relatively conserved, while the CI gene exhibits low variability in the SCMV subgroup, a subset of the genus *Potyvirus* [25,26,27]. Based on this evidence, both the CP and CI genes were selected as target genes for MDMV. In addition, we redesigned two primers with significant fragment size differences (greater than 100 bp) but similar annealing temperatures to optimize reaction conditions [28,29].

Ultimately, we established a multi-gene combined RT-PCR assay. This assay demonstrated high detection sensitivity, capable of detecting MDMV RNA at a concentration as low as 5.37 pg/μL, which is comparable to the detection limit of conventional RT-PCR. The assay also exhibited excellent specificity: When MDMV-positive samples were tested alongside positive samples of eight other viruses commonly found in maize and a blank control, only the MDMV-positive samples showed specific amplification, with no amplification observed in other samples or the blank control. In practical applications, the sequence identity between the detected positive samples and MDMV sequences on NCBI exceeded 93%. Moreover, the multi-gene combined RT-PCR assay established in this study showed greater accuracy than the commonly used RT-PCR method in practical sample testing, although its reaction time was slightly longer than that of conventional PCR. These findings highlight the significant application potential of this assay in virus detection in port quarantine and field production. Additionally, in this study, we optimized the quantities of CP gene and CI gene primers for MDMV amplification by increasing the amount of CI gene primers to reduce potential competition between the two genes during amplification. The underlying mechanism of this inhibitory effect warrants further investigation. When testing samples potentially carrying MDMV, the simultaneous detection of both conserved genes confirms the presence of MDMV. A sample is considered positive for MDMV if either or both of the CP and CI genes of MDMV are detected. If only one gene is detected, verification can be achieved by sequencing the PCR product or using RT-qPCR.

To verify the accuracy of multi-gene combined RT-PCR assay, this study developed an RT-qPCR assay for MDMV detection. Utilizing its quantitative analysis capabilities, the assay facilitated the construction of a standard curve for MDMV, enabling precise quantification of MDMV concentrations in carrying samples. The method demonstrated high sensitivity, detecting RNA concentrations as low as 0.537 pg/μL, and excellent specificity by distinguishing MDMV from eight other viruses. Additionally, the reaction time was significantly reduced.

In summary, the multi-gene combined RT-PCR assay developed in this study exhibits excellent specificity, sensitivity, and repeatability. This method enables the rapid and accurate detection of MDMV, offering valuable technical support for future quarantine initiatives and practical applications in MDMV management. Furthermore, there is currently a lack of research on plant virus detection using multi-gene combined RT-PCR methods. Looking ahead, combining multi-gene detection technology with RT-qPCR could facilitate the development of a multi-gene combined RT-qPCR assay. This would not only further improve the sensitivity of the assay but also reduce the time required for detection, thus providing a new option for the rapid detection of MDMV.

## Figures and Tables

**Figure 1 viruses-17-00370-f001:**
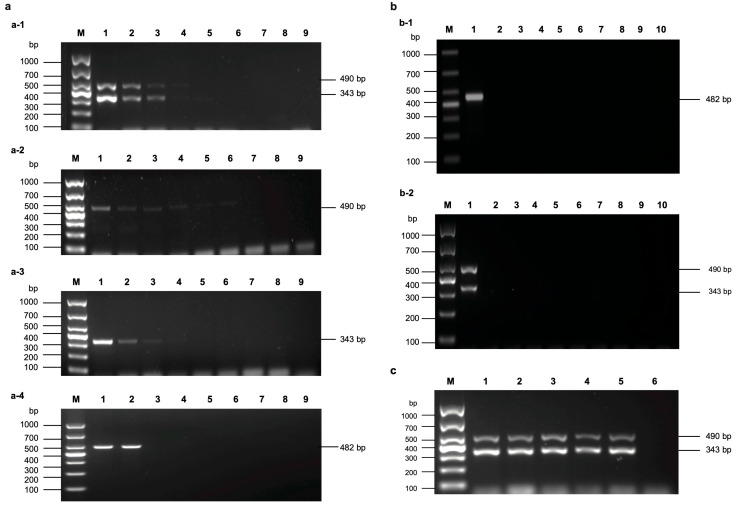
(**a**) Sensitivity of the multi-gene combined RT-PCR assay. (**a-1**) Multi-gene combined test; (**a-2**) PCR detection based on CI; (**a-3**) PCR detection based on CP; (**a-4**) RT-PCR assay. Lane M: Maker, Genstar Maker 1000 bp; Lane 1–8: 10^0^, 10^−1^, 10^−2^, 10^−3^, 10^−4^, 10^−5^, 10^−6^, and 10^−7^; Lane 9: negative control. (**b**) Specificity of the multi-gene combined RT-PCR assay. (**b-1**) RT-PCR assay (**b-2**) multi-gene RT-PCR assay. Lane M: Maker DNA, Genstar Maker 1000 bp; Lane 1: MDMV; Lane 2–9: SCMV, JGMV, WSMV, MCMV, BSMV, TMV, SRBSDV, and CMV; Lane 10: negative control. (**c**) Repeatability of the multi-gene combined RT-PCR assay. Lane M: Maker DNA, Genstar Maker 1000 bp; Lane 1–5: MDMV-positive samples; Lane 6: negative control.

**Figure 2 viruses-17-00370-f002:**
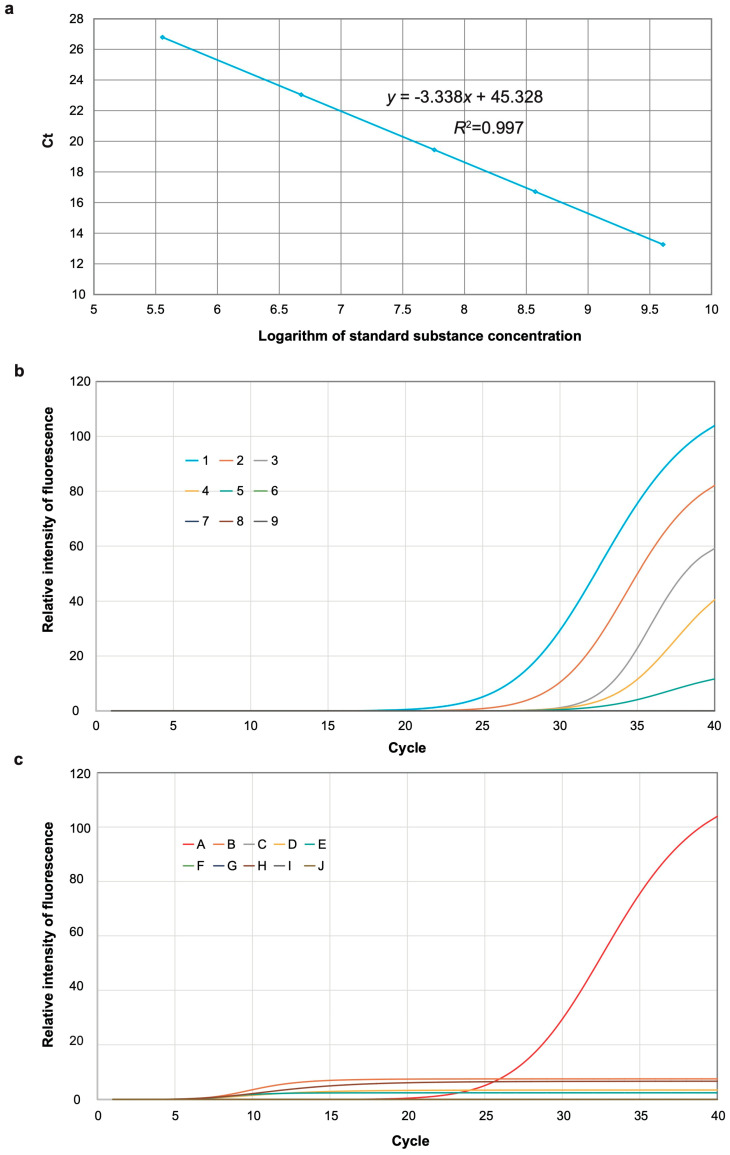
(**a**) Standard curve of RT-qPCR. (**b**) Sensitivity of RT-qPCR. 1–8: 10^−1^, 10^−2^, 10^−3^, 10^−4^, 10^−5^, 10^−6^, 10^−7^, and 10^−8^; 9: negative control. (**c**) Specificity of RT-qPCR. A: MDMV; B–I: SCMV, JGMV, WSMV, MCMV, BSMV, TMV, SRBSDV, and CMV; J: negative control.

**Table 1 viruses-17-00370-t001:** Primers and probes used in the multi-gene combined RT-PCR assay of maize. Samples and real-time fluorescent quantitative PCR detection in this study.

Gene	Primer Sequence (5′–3′)	Length (nt)	Tm (°C)	Expected Size (bp)
CP	MDMV482-F [23]: YGCATCTCCAACTTTCAGACA	23	56.7	482
	MDMV482-R [23]: CCTCACTCACTTGCVGACA	23	57.1	
	MDMV-CP-343-F: CCGAACATCAATGGTGTCTGG	21	56.4	343
	MDMV-CP-343-R: CGACATTCCCATCAAGACCAAAC	23	53.9	
	MDMV-CP-322-F: GAGCAACCAGGGCTGAATTTG	21	57.1	322
	MDMV-CP-322-R: CATAACGTGCAAGGCTAAAGTCG	23	56.4	
	MDMV-CP-F: GCCACAACAACAAGACTTATCGAA	24	62.5	75
	MDMV-CP-R: TTCTGCACTGCTTCATACCATCTAT	25	61.3	
	MDMV-CP-P:(FAM)ACCCGAGCAACCAGGGCTGAATTC(TAMRA)	24	72.1	/
CI	MDMV-CI-490-F: CCAATGGTGGTCAAATCAACTTGAG	25	56.3	490
	MDMV-CI-490-R: CTTCCCATTAAACGCATATTCTTTGAG	27	56.3	
	MDMV-CI-640-F: GCCAGGACTATGATGCAATTTGAGC	25	58.7	640
	MDMV-CI-640-R: CCAAATTCAAGCAAGTCCTCTGG	23	56.4	

Y = C/T, V = G/A/C.

**Table 2 viruses-17-00370-t002:** Repeatability test of the RT-qPCR assay for MDMV.

RNA Concentration	Ct Value	Mean Ct Value	Standard Deviation	Coefficient of Variation
53.7 ng/μL × 10^−1^	13.49	13.26	0.3658	2.76%
	12.98			
	13.31			
53.7 ng/μL × 10^−2^	17.19	16.71	0.5921	3.54%
	16.52			
	16.42			
53.7 ng/μL × 10^−3^	17.19	19.44	0.8125	4.18%
	16.52			
	16.42			
53.7 ng/μL × 10^−4^	23.46	23.04	0.5267	2.29%
	22.91			
	22.75			
53.7 ng/μL × 10^−5^	26.59	26.79	0.5396	2.01%
	27.23			
	26.55			

## Data Availability

All sequences of MDMV used in this study are publicly available on NCBI. The accession numbers used can be found in Appendix A.

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
