# Peer review of "A Novel Multi-Gene Combined RT-PCR Assay for Rapid and Sensitive Detection of Maize Dwarf Mosaic Virus"

_viruses, 2025, doi:10.3390/v17030370_

Round 1

Reviewer 1 Report

Comments and Suggestions for Authors

The manuscript viruses-3374826 describes a methodology to detect by RT-qPCR and other methods the potyvirus MDMV. The results of the manuscript are sound, but in my opinion, the manuscript requires further reorganization of the results and figures. In my opinion, there are too many figures and supplementary and some of the information could be integrated in the text and highlight the most important information. Also, most of the figures/tables and figure legends are not described well and more information should be provided. Thus, there are some ammendments that require attemption before accepting for publication in Viruses.

General and specific comments:

1) Figures and supplementary figures. I find very little information provided in the figures and supplementary figures. The electrophoresis gels on the supplementary files do not indicate what is loaded in each lane. Also, the authors should specify if PCR has been performed (with plasmids) or RT-PCR in each case. What are the negative controls on RT-PCR and PCRs in all figures, please specify (non-template control, healthy plant?). Did the authors include a non-template sample and a healthy plant?. Indicate in each figure which primers were used.

Also, I think figures 1-3 and 4-6 could be merge into two figures with different pannels since they provide the optimization of the same technique. I find there are too many figures.

2) Figure 4: please, specify what is the X logarithm with specific information about MDMV, is it a plasmid, RNA dilutions? And provide more information in the legend. In my opinion, this figure could be removed, but if you keep them provide the equation of the curve, the R2 value and efficiency (check other papers about RT-qPCR). If you decide to eliminate the figure, provide this information in the text.

3) Figure 5: Indicate the template for the dilutions and the nature of the negative control.

4) Positive samples: in general indicate in all the figures/tables and supplementary the nature of these samples. Are they plasmids, RNA, samples from field??. Indicate number if samples are the same used in Supplementary table 3.

5) Specificity of the tests: the authors provide specificity about different viruses that infect maize, but they do not show if the method can be used with isolates of MDMV of different diversity. ) I think it would be very important that the authors indicate in the text the high amount of samples tested and if they were genetically different. E.g.: do they know if these samples have high or low variability of the two regions tested?. Over 60 samples tested only 8 were positive. Did the authors sequenced those samples?

I think it would be important to show the variability of primers/probes within MDMV isolates, and I would like to request an alignment of all available sequences (or representative samples if too many to show in a figure) indicating how conserved are primers and probes and if this test can be used with isolates with different variability. I know there is a line where the authors state that the sequence similarity between positive samples and the MDMV sequence available on NCBI reached 97.65% (indicate the accession number in the text, line 324). Please, check other papers about RT-qPCR where they show alignments of primers and probes with available sequences and provide a similar supplementary figure.

6) Table 1: I would eliminate this table and replace it by the table of primers used in this work. It is important information since are part of the results. And regarding Table 1 just indicate that different dosages were tested ranging from X to X and indicate the dosage that was finally used.

7) Copy number of standard curve is not provided. The authors should consider the copy number used having in account the size of the template and the concentration used. Please, calculate the number of copies and provide the standard curve with the exact amounts of copy number of plasmid from 10-1 to 10-10 copies.

8) Supplementary Figure 1: indication of which sample is loaded in each lane is missing. How many samples were tested? Also, the name of primers is missing in the legend.

9) Table S2: Indicate genomic region in a column. GenBank accession used to design the primers and if degenerated primers indicate the meaning of each degenerate base. Change " Upstream and downstream primer sequence " to primer and probe sequence. Divide in 2 columns primer name and primer sequence and indicate 5'-3' .

10) Table S3: Indicate below the table: Meaning of +, - and 1,2,3. (The copy number would be much better indication).

11) About the primers used for multiplex, they are degenerated. Indicate how they were designed and based on which sequences.

Author Response

Reviewer # 1

The manuscript viruses-3374826 describes a methodology to detect by RT-qPCR and other methods the potyvirus MDMV. The results of the manuscript are sound, but in my opinion, the manuscript requires further reorganization of the results and figures. In my opinion, there are too many figures and supplementary and some of the information could be integrated in the text and highlight the most important information. Also, most of the figures/tables and figure legends are not described well and more information should be provided. Thus, there are some amendments that require attention before accepting for publication in Viruses.

RESPONSE: We sincerely thank the reviewer for taking the time to review our manuscript and for providing valuable feedback. We have incorporated the suggested revisions, as detailed in our point-by-point responses below.

General and specific comments:

1) Figures and supplementary figures. I find very little information provided in the figures and supplementary figures. The electrophoresis gels on the supplementary files do not indicate what is loaded in each lane. Also, the authors should specify if PCR has been performed (with plasmids) or RT-PCR in each case. What are the negative controls on RT-PCR and PCRs in all figures, please specify (non-template control, healthy plant?). Did the authors include a non-template sample and a healthy plant? Indicate in each figure which primers were used.

RESPONSE: We thank the reviewers pointed this out. As suggested by the reviewer, we have now added the information in the revision.

Also, I think figures 1-3 and 4-6 could be merge into two figures with different pannels since they provide the optimization of the same technique. I find there are too many figures.

RESPONSE: Merged as suggested.

2) Figure 4: please, specify what is the X logarithm with specific information about MDMV, is it a plasmid, RNA dilutions? And provide more information in the legend. In my opinion, this figure could be removed, but if you keep them provide the equation of the curve, the R2 value and efficiency (check other papers about RT-qPCR). If you decide to eliminate the figure, provide this information in the text.

RESPONSE: Removed as suggested.

3) Figure 5: Indicate the template for the dilutions and the nature of the negative control.

RESPONSE: A tenfold serial dilution of cDNA, originating from a positive sample containing MDMV, served as the template. This process yielded dilutions spanning from 10⁻¹ to 10⁻⁸. For negative controls, the cDNA was replaced with ddH₂O to ensure the validity of the experimental setup.

4) Positive samples: in general indicate in all the figures/tables and supplementary the nature of these samples. Are they plasmids, RNA, samples from field?? Indicate number if samples are the same used in Supplementary table 3.

RESPONSE: Indicated as suggested by the reviewer.

5) Specificity of the tests: the authors provide specificity about different viruses that infect maize, but they do not show if the method can be used with isolates of MDMV of different diversity). I think it would be very important that the authors indicate in the text the high amount of samples tested and if they were genetically different. E.g.: do they know if these samples have high or low variability of the two regions tested? Over 60 samples tested only 8 were positive. Did the authors sequence those samples?

RESPONSE: We appreciate the reviewer’s concern regarding the diversity of various MDMV isolates. We would like to clarify that the method we propose is applicable to different MDMV isolates, as the primers were designed based on highly conserved regions of the viral genome. In addition, degenerate bases were incorporated to accommodate highly variable sites, as detailed in Table 1 of the revised manuscript. Regarding the 8 positive samples identified out of the 60 tested, sequencing was conducted to confirm their results.

I think it would be important to show the variability of primers/probes within MDMV isolates, and I would like to request an alignment of all available sequences (or representative samples if too many to show in a figure) indicating how conserved are primers and probes and if this test can be used with isolates with different variability. I know there is a line where the authors state that the sequence similarity between positive samples and the MDMV sequence available on NCBI reached 97.65% (indicate the accession number in the text, line 324). Please, check other papers about RT-qPCR where they show alignments of primers and probes with available sequences and provide a similar supplementary figure.

RESPONSE: We thank the reviewer for this valuable suggestion. As suggested, we have included a supplementary figure in the revised manuscript that illustrate the alignments of primers and probes with available sequences.

6) Table 1: I would eliminate this table and replace it by the table of primers used in this work. It is important information since are part of the results. And regarding Table 1 just indicate that different dosages were tested ranging from X to X and indicate the dosage that was finally used.

RESPONSE: We thank the reviewer for this suggestion and have replaced this table with the Supplementary Table S1 from our initial submission.

7) Copy number of standard curve is not provided. The authors should consider the copy number used having in account the size of the template and the concentration used. Please, calculate the number of copies and provide the standard curve with the exact amounts of copy number of plasmid from 10-1 to 10-10 copies.

RESPONSE: The standard curve was constructed using 9 plasmid standards diluted with a ten-fold gradient of 4.3×109 copies/µL to 4.3×10 copies/µL. Using five dilutive plasmid standards -- 4.3×10⁵ copies/µL, 4.3×106 copies/µL, 4.3×107 copies/µL, 4.3×108 copies/µL, and 4.3×109 copies/µL construct the standard curve.

8) Supplementary Figure 1: indication of which sample is loaded in each lane is missing. How many samples were tested? Also, the name of primers is missing in the legend.

RESPONSE: In the revised manuscript, we had added further details on Supplementary Figure 1.

9) Table S2: Indicate genomic region in a column. GenBank accession used to design the primers and if degenerated primers indicate the meaning of each degenerate base. Change " Upstream and downstream primer sequence " to primer and probe sequence. Divide in 2 columns primer name and primer sequence and indicate 5'-3' .

RESPONSE: We have made revisions as suggested by the reviewer.

10) Table S3: Indicate below the table: Meaning of +, - and 1,2,3. (The copy number would be much better indication).

RESPONSE: We have added explanations of these symbols and numbers to the Supplementary Table S3.

11) About the primers used for multiplex, they are degenerated. Indicate how they were designed and based on which sequences.

RESPONSE: We have added a brief description of the primer design process in the revision. In addition, a supplementary figure has been provided to illustrate the alignments of primers with the available sequences.

Comments on the Quality of English Language

The English isn't very good, but I'm not an English speaker

RESPONSE: We now have invited a native English speaker to thoroughly review the revised manuscript and carefully revised the text to ensure linguistic accuracy and clarity. 

Reviewer 2 Report

Comments and Suggestions for Authors

Manuscript Viruses-3374826 describes two (or three) new molecular detection methods for maize dwarf mosaic virus, which strongly affects maize and is a quarantine pest in China. While the idea seems interesting and useful for managing the introduction of this virus via export to China, the manuscript is sorely lacking in detail and scientific rigor. It also seems to have been written very quickly, with numerous errors of form and grammar.

Please find below my detailed comments, point by point.

Abstract: The abstract contains far too many errors. As it's the first text we read, it's important to take particular care in writing it.

- l. 15: Why is only the multi-gene test mentioned and not the RT-qPCR test, which is also mentioned in the manuscript?

- l. 16: “samples” and “other plants”. But barley is the only other plant, so you might as well write it down.

- l. 18: it's not the primers that have been improved, but the PCR test conditions.

- l. 18-19: there is no mention of screening primers in the paper.

- l. 23: the end of the sentence is incorrect. And the results presented here do not correspond to those described in the results section.

Introduction: The introduction is a little light. We're talking about a virus for which the authors have developed two new detection tests. It's important to develop this part more fully, by talking about what we know about the virus, whether detection tests already exist, etc.

- L. 31: a reference is requested

- L. 41: remove the “the” before “America”

- L. 48: a figure showing the symptoms of MDMV would be a good idea

- L. 58: remove the comma

- L. 67-71: only the RT-qPCR test is mentioned, even though it is described in the materials and methods and in the results. Why is this?

Materials and methods: this section lacks precision on some aspects, and on the contrary develops others too much (description of kits). It needs to be thoroughly revised.

- L. 74: remove “test”

- L. 74-75: how are samples taken, and in what form (leaves, I imagine?)? This needs to be clarified.

- L. 75-76: are they cultivated or just kept in the freezer?

- L. 77-80: Ditto for positive controls for other viruses.

- L. 80-82: How are the primers designed? From which sequence(s)? Give the GenBank number(s) and explain how the alignment is done, if necessary.

- L. 83: “The synthesis” and not “these synthesis”.

- L. 83: the S2 table should not be in supplementary material. For the development of a detection test, it is really very important.

- L. 88: what the leaf weight (if any)?

- L. 95: the volume of a product is useless if you don't know its concentration. It's better to express the quantity. What domain of the virus genome do these primers target? Have they already been published? If so, include the reference.

- L. 96: refer to table S2

- L. 104: refer to table S2.

- L. 109: Why does the annealing step take 90 sec? It seems like such a long time!

- L. 120: which sample?

- L. 133: positions must be set in table S2

- L. 140-143: which kit is used for cloning? which competent bacteria are used for transformation?

- L. 149: refer to table S2

- L. 157: which sample?

- L. 161: repeatability tests should have been carried out on a positive sample, or even several, rather than on purified plasmid, with dilutions using a negative sample. Otherwise, real detection conditions are not respected.

Results: It is not shown that the conventional test leads to false positives, although this is precisely what is announced in the introduction (L. 65). The comparison is unclear; it would have been better to draw up a summary table with samples tested with the 3 methods to show which is the most effective, rather than gel figures which don't add up to much in the end.

- L. 175: which sample?

- L. 183-184: what does “best amplification effect” mean? to be reworded.

- L. 188-194: which sample was tested to establish the PCR parameters?

- L. 201-205: how were dilutions made, from which sample?

- L. 206: compare with conventional RT-PCR.

- L. 211-212: which PCR products? compared with which sequence?

- L. 226: how are dilutions made, from what?

- L. 229: how is this limit calculated?

- L. 242-250: the comparative results of the 3 tests should be presented on a table rather than on gel photos.

- L. 251-254: more precision is needed: which PCR products were sequenced, how, what is the percentage of homology with the GenBank sequence? Phylogeny would have been a plus to try and understand why this sequence is detected by the multi-gene test and RT-qPCR and not by conventional RT-PCR.

- Figure 4 and figure 5: useless

Discussion: in what way are the new tests developed really more sensitive and more effective than the conventional method? With 60 samples tested and only 8 positive results, we can't really see the added value of this new test, especially as it has not been demonstrated that the conventional test leads to false positives, as stated in the introduction. In the end, it's hard to understand which test to choose, how to use them and under what conditions.

Supplementary data:

-        Figure S1: the legend is missing

-        Figure S2: the legend is also missing. Nowhere does it say why there are 4 band sizes and not two, as expected. I assume that you tested several primer pairs, as announced in the introduction, but this screening is not described in the materials and methods section or in the results section. So why did you choose this primer pair rather than the other? And what about the other combinations?

-        Figure S3: still no legend

-        Figure S4: ditto

-        Figure S5 : a comparison table would be much better

-        Figure S6 : same as figure S5

-        Table S1: There's an error in the sample listing: the numbers 15 and 16, and 45 and 46 are found twice. The two batches should be separated by a line, for greater clarity.

-        Table S2: this table is almost the most important in the entire manuscript, and should not be included as supplementary material. However, it lacks an essential piece of information: the position of the primers on the virus genome. Next, it lists two primer pairs never mentioned in the manuscript (MDMV-CP-322-F/R and MDMV-CI-640-F/R).

-        Table S3: noted S4 in the title!! should be combined with indexing results with the multi-gene assay and conventional RT-PCR, rather than gel photos.

Comments on the Quality of English Language

The English isn't very good, but I'm not an English speaker

Author Response

Reviewer #2

General Comments

Manuscript Viruses-3374826 describes two (or three) new molecular detection methods for maize dwarf mosaic virus, which strongly affects maize and is a quarantine pest in China. While the idea seems interesting and useful for managing the introduction of this virus via export to China, the manuscript is sorely lacking in detail and scientific rigor. It also seems to have been written very quickly, with numerous errors of form and grammar.

RESPONSE: We sincerely thank the reviewer for taking the time to review our manuscript and for providing valuable feedback. Your constructive comments have been invaluable in improving the quality and rigor of our work. We have incorporated the suggested revisions into the revised manuscript and have emphasized the practical applications of our detection methods, particularly their potential impact on quarantine measures and virus management strategies. We apologize for the errors in form and grammar in our initial submission. To address this, we have invited a native English speaker to thoroughly review the revised manuscript. Additionally, we have carefully revised the text to ensure linguistic accuracy and clarity. Please see our point-by-point responses below.

Specific comments

Please find below my detailed comments, point by point.

Abstract: The abstract contains far too many errors. As it's the first text we read, it's important to take particular care in writing it.

RESPONSE: We apologize for the errors in the abstract of our initial submission. In the revised manuscript, we have carefully rephrased the abstract to ensure clarity, accuracy, and coherence.

- l. 15: Why is only the multi-gene test mentioned and not the RT-qPCR test, which is also mentioned in the manuscript?

RESPONSE: RT-qPCR is primarily used as a verification method for multi-gene combined detection, and its reaction conditions and system have not been fully optimized. Therefore, in the initial manuscript, we only briefly mentioned this method without providing a detailed description. However, as suggested, we have now expanded the discussion and included additional details about RT-qPCR in the revised manuscript.

- l. 16: “samples” and “other plants”. But barley is the only other plant, so you might as well write it down.

RESPONSE: Revised as suggested.

- l. 18: it's not the primers that have been improved, but the PCR test conditions.

RESPONSE: We adjusted and optimized the ratio of primer concentrations in the multi-gene combined detection aasay to determine the optimal primer amounts. Additionally, the annealing temperature of the PCR reaction was refined to improve performance.

- l. 18-19: there is no mention of screening primers in the paper.

RESPONSE: Added as suggested.

- l. 23: the end of the sentence is incorrect. And the results presented here do not correspond to those described in the results section.

RESPONSE: Revised as suggested.

Introduction: The introduction is a little light. We're talking about a virus for which the authors have developed two new detection tests. It's important to develop this part more fully, by talking about what we know about the virus, whether detection tests already exist, etc.

RESPONSE: We have made revisions as suggested by the reviewer.

- L. 31: a reference is requested

RESPONSE: We have added a relevant reference to our revised manuscript.

- L. 41: remove the “the” before “America”

RESPONSE: Removed as suggested.

- L. 48: a figure showing the symptoms of MDMV would be a good idea

RESPONSE: We thank the reviewer for this suggestion. Unfortunately, we did not have access to plant leaves or other symptomatic materials infected with MDMV, as our samples were collected from maize, sorghum, and barley seeds intended for entry quarantine at the customs port.

- L. 58: remove the comma

RESPONSE: Removed as suggested.

- L. 67-71: only the RT-qPCR test is mentioned, even though it is described in the materials and methods and in the results. Why is this?

RESPONSE: Added this part as suggested.

Materials and methods: this section lacks precision on some aspects, and on the contrary develops others too much (description of kits). It needs to be thoroughly revised.

RESPONSE: We have made revisions as suggested by the reviewer.

- L. 74: remove “test”

RESPONSE: Removed as suggested.

- L. 74-75: how are samples taken, and in what form (leaves, I imagine?)? This needs to be clarified.

RESPONSE: The samples were the seeds of maize, sorghum, and barley imported from customs ports. The sentence has been rephrased t for clarify.

- L. 75-76: are they cultivated or just kept in the freezer?

RESPONSE: The samples were the seeds of maize, sorghum, and barley imported from customs ports.

- L. 77-80: Ditto for positive controls for other viruses.

RESPONSE: The positive controls used were synthetic positive samples purchased commercially. We’ve added a short explanation to clarify their use.

- L. 80-82: How are the primers designed? From which sequence(s)? Give the GenBank number(s) and explain how the alignment is done, if necessary.

RESPONSE: We thank the reviewer for point this out. As suggested, we have added a brief description of the primer design process and included a supplementary figure in the revised manuscript that illustrate the alignments of primers and probes with available sequences.

- L. 83: “The synthesis” and not “these synthesis”.

RESPONSE: Revised as suggested.

- L. 83: the S2 table should not be in supplementary material. For the development of a detection test, it is really very important.

- L. 88: what the leaf weight (if any)?

RESPONSE: The sample was not a leaf; instead, we added 0.1g of ground crop seed powder to extract RNA.

- L. 95: the volume of a product is useless if you don't know its concentration. It's better to express the quantity. What domain of the virus genome does these primers?

RESPONSE: All the primers used in the experiment were prepared at a concentration of 10 U/μL. As suggested, we have added a brief description of these primers. Specifically, the primers MDMV-482-F and MDMV-482-R were designed from the conserved domain of the CP region. This information has been included in Table 1 of the revised manuscript.

- L. 96: refer to table S2

RESPONSE: Refereed as suggested.

- L. 104: refer to table S2.

RESPONSE: Refereed as suggested.

- L. 109: Why does the annealing step take 90 sec? It seems like such a long time!

RESPONSE: The annealing step duration of 90 seconds was chosen in accordance with the protocol outlined in the kit’s instructions. This timing is specifically recommended to ensure optimal results under the prescribed conditions.

- L. 120: which sample?

RESPONSE: This sample is the maize seed sample infected with MDMV that entered Fuzhou port.

- L. 133: positions must be set in table S2

RESPONSE: Set as suggested.

- L. 140-143: which kit is used for cloning? which competent bacteria are used for transformation?

RESPONSE: The kit used for cloning is Ultra-Universal TOPO cloning kit, the competent bacteria is Escherichia coli Mach1 T1R strain. We had added brief description on the kit for cloning and the bacteria used for transformation in the revised manuscript.

- L. 149: refer to table S2

RESPONSE: Refereed as suggested.

- L. 157: which sample?

RESPONSE: Refereed as suggested.

- L. 161: repeatability tests should have been carried out on a positive sample, or even several, rather than on purified plasmid, with dilutions using a negative sample. Otherwise, real detection conditions are not respected.

RESPONSE: Thanks for your suggestion, we have replaced the template of the repeatability test with the sample cDNA infected with MDMV.

Results: It is not shown that the conventional test leads to false positives, although this is precisely what is announced in the introduction (L. 65). The comparison is unclear; it would have been better to draw up a summary table with samples tested with the 3 methods to show which is the most effective, rather than gel figures which don't add up to much in the end.

RESPONSE: Traditional virus detection methods, such as DAS-ELISA, have the limitation of low sensitivity when detecting viruses in seeds, which may lead to false negative results. Similarly, conventional RT-PCR methods based on a specific gene may also yield false negatives if there are significant variations in the gene segment. We have revised this section accordingly and, as suggested, summarized the test results for the three methods.

- L. 175: which sample?

RESPONSE: The sample refers to the maize seed sample infected with MDMV that was imported through Fuzhou Port.

- L. 183-184: what does “best amplification effect” mean? to be reworded.

RESPONSE: Reworded as suggested.

- L. 188-194: which sample was tested to establish the PCR parameters?

RESPONSE: The sample used to establish the PCR parameters was a maize seed sample infected with MDMV, which was imported through Fuzhou Port.

- L. 201-205: how were dilutions made, from which sample?

RESPONSE: The dilutions were prepared in a tenfold gradient using the maize seed sample infected with MDMV, which was imported through Fuzhou Port.

- L. 206: compare with conventional RT-PCR.

RESPONSE: Compared as suggested.

- L. 211-212: which PCR products? compared with which sequence?

RESPONSE: Sequencing of all PCR products obtained from the multi-gene combined detection assay revealed that the sequences shared more than 94% similarity with the reported MDMV sequences. In the revised manuscript, we have rephrased the sentence to enhance clarity and provide more specific details.

- L. 226: how are dilutions made, from what?

RESPONSE: The samples were diluted in a tenfold gradient. 

- L. 229: how is this limit calculated?

RESPONSE: The limit was calculated based on the RT-qPCR amplification curve. Amplification was observed up to the dilution of 10-5, corresponding to a concentration is 53.7 pg/μg RNA.

- Figure 4 and figure 5: useless

RESPONSE: In revised manuscript, Figure 4 has been deleted, and Figure 5 has been merged with Figure 6.

Discussion: in what way are the new tests developed really more sensitive and more effective than the conventional method? With 60 samples tested and only 8 positive results, we can't really see the added value of this new test, especially as it has not been demonstrated that the conventional test leads to false positives, as stated in the introduction. In the end, it's hard to understand which test to choose, how to use them and under what conditions.

RESPONSE: The newly developed method demonstrates a higher detection rate (13.3%) for imported samples compared to the conventional method (11.6%).  In the revised manuscript, we have provided additional details regarding the contexts in which the new test is most effective. Moreover, the detection results of the new method are more accurate, as it minimizes the risk of false positives and provides reliable results for practical applications. While the conventional method has not been explicitly shown to lead to false positives in this study, the multi-gene detection assay improves specificity and accuracy by targeting two conserved genes of MDMV simultaneously. This reduces the likelihood of ambiguous results, especially when dealing with complex or mixed virus infections.

Supplementary data:

-  Figure S1: the legend is missing

RESPONSE: Added as suggested.

-  Figure S2: the legend is also missing. Nowhere does it say why there are 4 band sizes and not two, as expected. I assume that you tested several primer pairs, as announced in the introduction, but this screening is not described in the materials and methods section or in the results section. So why did you choose this primer pair rather than the other? And what about the other combinations?

RESPONSE: As suggested, we have added legend for Figure S2 (now presented as Figure S3 in the revision) and included a section on primer screening in the manuscript. Ultimately, we selected the primer set that demonstrated the best amplification results after testing various combinations.

-  Figure S3: still no legend

RESPONSE: Added as suggested.

-  Figure S4: ditto

RESPONSE: Added as suggested.

-  Figure S5: a comparison table would be much better

RESPONSE: As suggested by the reviewer, we have now added a comparison table, now presented as Table S3.

-  Figure S6: same as figure S5

RESPONSE: As suggested by the reviewer, we have now added a comparison table, now presented as Table S3.

-  Table S1: There's an error in the sample listing: the numbers 15 and 16, and 45 and 46 are found twice. The two batches should be separated by a line, for greater clarity.

RESPONSE: Our mistake. Revised as suggested.

-  Table S2: this table is almost the most important in the entire manuscript, and should not be included as supplementary material. However, it lacks an essential piece of information: the position of the primers on the virus genome. Next, it lists two primer pairs never mentioned in the manuscript (MDMV-CP-322-F/R and MDMV-CI-640-F/R).

RESPONSE: As suggested by the reviewer, Table S2 has been replaced and is now presented as Table 1. In addition, the screening procedures for MDMV-CP-322-F/R and MDMV-CI-640-F/R have been included into the revised manuscript.

-  Table S3: noted S4 in the title!! should be combined with indexing results with the multi-gene assay and conventional RT-PCR, rather than gel photos.

RESPONSE: We appreciate the reviewers for bringing this to our attention. In the revised manuscript, we have corrected the table name and incorporated the results of the multi-gene detection assay with RT-PCR into it.

Reviewer 3 Report

Comments and Suggestions for Authors

Good paper requiring minor address of comments! 

Please refer to the attached comments!

Author Response

Reviewer #3

Line 16: “Samples” and not “sampled”

Is it “corn” or “maize”??? The title of manuscript indicates maize so you need to be consistent with the term to use throughout the manuscript

Target crop is maize so remove wheat, sorghum, barley and others unless MDMV infects these other crops as well then you need to state in the introduction with supporting references. In this case, then you can keep the other crops.

RESPONSE: Changed as suggested.

Line 23: Delete “and” before the word “respectively”

RESPONSE: Deleted as suggested.

Line 47: How does MDMV differentiate from other similar and close maize viruses in terms of symptoms?

RESPONSE: MDMV symptoms, such as mosaic leaf patterns, stunted growth, and overall maize decline, are nearly identical to those caused by related viruses. As these symptoms overlap significantly, accurate identification requires supplementary diagnostic methods (i.e. molecular techniques), as visual inspection alone is insufficient.

Line 74: Where these grain or leaf samples??? Indicate please!

I am confused with the title of the manuscript and the content flow!!! Can you recast the title!!

It seems the work is “Development of a multigene detection technology for MDMV targeting Maize Imports” and not the one you have!! Recast!

RESPONSE: We thank the reviewer for pointing this out. The samples were the seeds of maize, sorghum, and barley imported from customs ports. As suggested by the reviewer, we have revised the title of the manuscript.

Line 83: Indicate the method used for the synthesis of the primers and probes with supporting references!

RESPONSE: The primers were synthesized by Shanghai Sangon Biological Engineering Technology and Service Co., Ltd. (Shanghai, China). In the revised manuscript, we have included a brief description of the synthesis method.

Line 87: Which type of instrument was used to run PCR?

RESPONSE: Biometra Tadvanced 96SG was used in this experiment to run PCR.

Line 311: I have not seen this analysis of PVY and SCMV genomes subgroup in the manuscript?

RESPONSE: we have clarified the relevant sections and rephrased the sentences to eliminate any potential confusion.

Line 317-322: Could you support this statement with some references if any??

RESPONSE: This statement is supported in the subsection of primer design within Materials and Methods section.

Line 325-327: Where else has this phenomenon occurred? Support it with references if any!!

RESPONSE: In the revised manuscript, we have rephrased the sentences to avoid the potential confusion.

Line 370: Where did you get the funding for this study? Recheck the statement you put up here!

RESPONSE: Our mistakes. The statement has been corrected in our revised manuscript.

Round 2

Reviewer 1 Report

Comments and Suggestions for Authors

Thanks to the authors for the effort making corrections. I have just some minor comments/edits:

Supplementary Fig. 1: check the names of the primers in the arrows. I believe there are some duplications and mistakes. E.g: MDMV-CP-P is missing in one of the arrows. Alignment for MDMV-CI-490-F/MDMV-CI-490-R is missing and duplicated by MDMV-CI-640-F/MDMV-CI-640-R. Check this figure.

Legend of Table S2 says in the MS: Table S2. Primers of different multiples but the name in the supplementary file is different, please change it to the correct one.

Author Response

Comments
Thanks to the authors for the effort making corrections. I have just some minor comments/edits:

Supplementary Fig. 1: check the names of the primers in the arrows. I believe there are some duplications and mistakes. E.g: MDMV-CP-P is missing in one of the arrows. Alignment for MDMV-CI-490-F/MDMV-CI-490-R is missing and duplicated by MDMV-CI-640-F/MDMV-CI-640-R. Check this figure.

RESPONSE: Checked and Changed as suggested.

Legend of Table S2 says in the MS: Table S2. Primers of different multiples but the name in the supplementary file is different, please change it to the correct one.

RESPONSE: Checked and Changed as suggested.

Reviewer 2 Report

Comments and Suggestions for Authors

The authors have made many corrections to the previous version, as well as clarifying their sampling and implementation, as requested. However, there are still many errors and, above all, an overall meaning to this work that I still haven't grasped. Indeed, what is really the major interest of this new multi-gene detection method for MDMV compared with other existing method or developed in parallel? Increased specificity vis-à-vis other potyviruses is the main argument in favor of this new method. However, the other methods (conventional RT-PCR and RT-qPCR) ultimately show the same specificity. Better sensitivity? qPCR is 100 times more sensitive than the multi-gene test. I therefore think that the angle used to present these detection techniques is not the right one, and the paper deserves to be totally revised for greater consistency.

Here are my comments, point by point:

- L. 135: give the GenBank accession number of the genome used as reference for the alignment. Moreover, in figure S1, the sequences of the alignment are presented under an accession number, probably from GenBank. It should be explained where these sequences come from, and if any of the sequences analyzed in this article have been deposited on GenBank, this should be specified.

- L. 136: “genes” and not “gens”.

- L. 139: “only” is missing just before “from the CP gene”.

- L. 178: probably “starting” and not “strating”.

- L. 344-345: I don't really understand how the primer pair chosen gives better amplification results than the other, according to the gel photo in figure S3.

- L. 417: not “b”, but “c”.

- L. 451: RT-qPCR is 100X more sensitive than the multi-gene method and just as specific. Why not make the qPCR method the choice for routine use? This is not explained, and I think it’s an essential point.

- L. 549: I only see 6 signals, not 7.

- L. 551: how many PCR products were sequenced? Only one (according to the results announced by the authors), two (according to my interpretation of the gels), more, but if so, which ones? This really lacks precision.

- L. 576-577: there's no real difference with the classic test in terms of specificity.

- L. 650-651: I don't see how the above paragraph shows that the multi-gene test is more reliable. In the end, I still don't really understand what this method offers compared to the other two.

- L. 676: the different primer dosages should be detailed.

- L. 741: reference should be made here to the sequences deposited on GenBank, if any.

Author Response

comments
The authors have made many corrections to the previous version, as well as clarifying their sampling and implementation, as requested. However, there are still many errors and, above all, an overall meaning to this work that I still haven't grasped. Indeed, what is really the major interest of this new multi-gene detection method for MDMV compared with other existing method or developed in parallel? Increased specificity vis-à-vis other potyviruses is the main argument in favor of this new method. However, the other methods (conventional RT-PCR and RT-qPCR) ultimately show the same specificity. Better sensitivity? qPCR is 100 times more sensitive than the multi-gene test. I therefore think that the angle used to present these detection techniques is not the right one, and the paper deserves to be totally revised for greater consistency.

RESPONSE: Thank you very much for your time in reviewing our manuscript and for your valuable comments. Mutations, genetic drift, and selective pressures can cause viruses, especially RNA viruses, to undergo various sequence variations. As we know, MDMV is an RNA virus with a high mutation rate. Since a single gene fragment constitutes only a portion of the virus's complete genome, significant mutations in this fragment may lead to inaccurate detection and false-negative results. Multi-gene combined detection, targeting two or more viral genes, effectively overcomes the issue of detection inaccuracies caused by single-gene fragment mutations, thereby offering higher accuracy and reliability. Multi-gene combined RT-PCR is a specialized form of multiplex RT-PCR, in which multiple pairs of primers are used simultaneously in a single reaction system to amplify different gene fragments of the same virus. Previous studies have demonstrated that the CP and CI genes in potyvirus genomes are relatively conserved. In this study, we focused on the CP and CI genes of MDMV to design and screen specific primers. After optimizing reaction systems and conditions, we established a rapid detection method capable of simultaneously detecting both target genes. The RT-qPCR method developed in this study aims to validate the practical applicability of the multi-gene combined detection approach for MDMV. As is well known, RT-qPCR exhibits higher sensitivity compared to RT-PCR. Therefore, this study utilized RT-qPCR to simultaneously detect MDMV in imported corn samples at ports, confirming the accuracy of the multi-gene combined detection method. Theoretically, multi-gene combined RT-PCR is more specific because it targets multiple genes simultaneously. In this study, to verify the specificity of the two pairs of primers in the multi-gene combined RT-PCR method, a specificity assay was performed against other viruses, belonging to both exclusive and inclusive genera that are capable of infecting corn crops. These reference viruses were also detected by conventional RT-PCR and RT-qPCR to determine the specificity of the conventional RT-PCR primers, RT-qPCR primers and probes.

To date, there have been no research reports on the detection of MDMV using a multi-gene combined RT-PCR assay. To avoid potential misunderstandings among readers, we have revised the title to ‘Rapid detection of maize dwarf mosaic virus using multi-gene combined RT-PCR assay’ in subsequent revision, along with corresponding updates to the abstract and main text.

Here are my comments, point by point:

- L. 135: give the GenBank accession number of the genome used as reference for the alignment. Moreover, in figure S1, the sequences of the alignment are presented under an accession number, probably from GenBank. It should be explained where these sequences come from, and if any of the sequences analyzed in this article have been deposited on GenBank, this should be specified.

RESPONSE: All available MDMV complete genome sequences were obtained from the NCBI GenBank database, as suggested. We have updated paragraph 2 of Chapter 2.1 to include that all available MDMV complete genome sequences were retrieved from the NCBI GenBank database.

- L. 136: “genes” and not “gens”.

RESPONSE: Changed as suggested.

- L. 139: “only” is missing just before “from the CP gene”.

RESPONSE: Added as suggested.

- L. 178: probably “starting” and not “strating”.

RESPONSE: Changed as suggested.

- L. 344-345: I don't really understand how the primer pair chosen gives better amplification results than the other, according to the gel photo in figure S3.

RESPONSE: Although the two primers MDMV-CP-343 and MDMV-CI-490 were not the best choice when they detect MDMV alone, they are the best primers combination for detecting MDMV. We found that the gel electrophoresis results of their amplified target fragment showed more similar brightness and were brighter overall than those of the other two primers.

- L. 417: not “b”, but “c”.

RESPONSE: Changed as suggested.

- L. 451: RT-qPCR is 100X more sensitive than the multi-gene method and just as specific. Why not make the qPCR method the choice for routine use? This is not explained, and I think it’s an essential point.

RESPONSE: Thank you very much for your professional advice. This study aims to establish a multi-gene combined RT-PCR assay for the rapid, specific, sensitive, and reliable detection of MDMV, without the need for special or expensive equipment. The RT-qPCR method developed in this study was only used to validate the effectiveness of the multi-gene combined RT-PCR assay. We have added a corresponding description in the Introduction section, stating that an RT-qPCR assay based on the CP gene was also developed to validate the practical application of the MDMV multi-gene combined RT-PCR assay in this study. In addition, the RT-qPCR assay is costlier compared to the multi-gene combined RT-PCR assay and requires expensive equipment.

- L. 549: I only see 6 signals, not 7.

RESPONSE: In figure S7(a), there is also a relatively weak signal on lane 6. The overall sensitivity of MDMV detected by conventional RT-PCR is lower, so the signal shown is weak. Thank you for pointing out that. We have adjusted the overall brightness of the gel electrophoretic map for convenience of viewing.

- L. 551: how many PCR products were sequenced? Only one (according to the results announced by the authors), two (according to my interpretation of the gels), more, but if so, which ones? This really lacks precision.

RESPONSE: We sequenced the PCR products of the 8 positive samples detected by the multi-gene combined RT-PCR test. We are sorry that we only generalized it before, but now we have compared the sequencing results with the NCBI database and displayed them in Table S4.

- L. 576-577: there's no real difference with the classic test in terms of specificity.

RESPONSE: Different viruses, belonging to both exclusive and inclusive genera capable of infecting corn crops, were used to verify the specificity of the primers for multi-gene combined RT-PCR, conventional RT-PCR primers, and probes for RT-qPCR, to demonstrate that all three methods are highly specific. The results above are not intended to compare the specificity of the three methods. Due to the limited number of viruses tested, it may not be possible to conclude which method is more specific.

- L. 650-651: I don't see how the above paragraph shows that the multi-gene test is more reliable. In the end, I still don't really understand what this method offers compared to the other two.

RESPONSE: Compared to the conventional RT-PCR method and RT-qPCR, which rely on a single gene, the multi-gene combined RT-PCR assay can detect both the relatively conserved CP and CI genes of MDMV in a single PCR experiment, thus preventing false-negative results due to mutations in a single gene fragment of MDMV. To avoid ambiguity, we have reworded the relevant paragraphs in the Discussion section.

- L. 676: the different primer dosages should be detailed.

RESPONSE: Detailed as suggested.

- L. 741: reference should be made here to the sequences deposited on GenBank, if any.

RESPONSE: Added as suggested.